# Unconditioned and learned morphine tolerance influence hippocampal-dependent short-term memory and the subjacent expression of GABA-A receptor alpha subunits

**Ghazaleh Ghamkharinejad**[☯], **Seyed Hossein Marashi**[ID][☯], **Forough Foolad**, **Mohammad Javan, Yaghoub Fathollahi**[ID]*

Department of Physiology, Faculty of Medical Sciences, Tarbiat Modares University, Tehran, Iran

☯ These authors contributed equally to this work.
* fatolahi@modares.ac.ir

**Data Availability Statement:** The data can be found on the G-Node database (https://gin.g-node.org/), with provided URLs https://gin.g-node.org/

## Abstract

### Background

ɣ-aminobutyric acid (*GABA*) facilitator valproic acid may be able to curb memory disruption induced by morphine exposure.

### Objective

The effects of the *GABA* facilitator valproic acid on the behavioral tolerance induced by morphine were investigated. Then hippocampal-dependent tasks named spatial-working and short-term memory procedures using the Y-maze apparatus were examined in morphine tolerant rats. Finally, the changes in the expression of hippocampal *GABA-A* receptors underlying morphine tolerance were also examined.

### Methods

Rats were treated with daily morphine injections, with or without distinct contextual pairing. To examine the effect of valproic acid on morphine tolerance expression, valproic acid was pretreated an hour before morphine. Spatial-working and short-term memory procedures using the Y-maze apparatus were examined in morphine tolerant rats. Afterwards the changes in the expression of hippocampal *GABAα* receptors using the quantitative real-time PCR and western blot techniques to detect *GABArα* subunits mRNAs and protein level were studied.

### Results

Our results showed that both learned and non-associative morphine tolerance influence short-term memory and the subjacent expression of *GABArα* mRNAs and protein level. Despite its attenuating effects on the development and expression of both learned and non-

Ghazaleh-Ghamkharinejad/PONE-D-20-29685R1/
commit/
5ec65191342c636b5394558b6fee3cfcce2fa8ae
https://gin.g-node.org/Ghazaleh-Ghamkharinejad/
PONE-D-20-29685R1/commit/
aa6d40475a973df08a7f315eaa5c5c94d94c4d71.

**Funding:** This study was supported by from Iran National Science Foundation (INSF- 96008684) and Med-from Tarbiat Modares University to Y.F (www.modares.ac.ir), and Iranian Council for Cognitive Sciences and Technologies to Gh.Gh. (www.cogc.ir).

**Competing interests:** The authors have declared that no competing interests exist.

associative morphine tolerance, only associative morphine tolerance-induced memory dysfunction was ameliorated by valproic acid pretreatment. We also found that the expression of *GABArα1*, *α2*, *α5* subunits mRNAs and *GABAα* protein level were affected heavier in associative morphine tolerant rats.

## Conclusion

Our data supports the hypothesis that unconditioned and learned morphine tolerance influences short-term memory and the expression of *GABArα 1*, *α2*, *α5* mRNAs and *GABArα* protein level differently, and adds to our understanding of the behavioral and molecular aspects of the learned tolerance to morphine effects.

## 1. Introduction

Opioids, specifically morphine, remain the mainstream prescription for perioperative and chronic pain suppression [1]. The increase in opioid prescriptions has been associated with a higher risk of addiction, opioid-related mortality, and analgesic tolerance [2]. Opioid administration may lead to several complex modifications on the cell level, including receptor phosphorylation, signaling, and trafficking, resulting in the alteration of neuronal MOR (Mu opioid receptors) function in both peripheral and central nervous systems [3–5]. Opiate tolerance involves both associative and non-associative changes. Prolonged or repeated intake of morphine leads to non-associative morphine tolerance (NAMT). In addition, since learning circuits can contribute to the development and expression of morphine tolerance, contextual pairing during morphine administration can induce tolerance specific to that particular setting, which is called associative morphine tolerance (AMT) [6]. Although some reports exist on the underlying mechanisms of opiate associative tolerance, the processes that promotes this type of tolerance remains unknown.

Through synaptic modification and several molecular mechanisms, morphine plays a key role in the performance of adult rats in cognitive tasks and the attenuation of long-term potentiation (LTP), one of the major cellular mechanisms that underlies learning and memory [7]. The hippocampal formation is a crucial area to mediate learning and memory and 20%–50% of *GABArs* (γ-aminobutyric acid (*GABAA*) receptors) were expressed at its inhibitory synapses. Co-assembly of α, β and γ subunits create the most common receptors in all of the hippocampal sub-regions [8]. The α1 subunit constitute the dominant interface where benzodiazepines bind and it is necessary for a rapid formation of active synaptic contacts and the synaptogenesis effect. This subunit has been shown to associate with learning deficit in epilepsy, stress resilience and mediates plasticity in context-dependent learning [9]. The α2 subunit is located on the axon initial segment (AIS) and is involved in the reduction of postsynaptic inhibitory output and IPSC (inhibitory postsynaptic current) amplitude, morphine express and acquisition-expression of morphine reward at the dorsal hippocampus [10–13]. It has been documented that the α5 subunit is primarily expresses extra-synaptic in dendritic fields of hippocampal pyramidal cells. It is unique as its specific impact on cognitive behavior through tonic inhibitory current generating and neural oscillations alteration during learning behavior [14]. On the other hand, it has been mentioned that valproic acid (VPA), an anticonvulsant and mood stabilizing drug, was used to attenuate some physiological aspects of addiction to opioids by enhancing the GABAergic system [15, 16].

With the issue of morphine tolerance being so widespread in clinical studies and considering the effects of this exogenous opioid on cognitive functions, it would be important to clarify how working and short-term memory are affected in this model. In addition, with regard to the numerous documents available on the involvement of GABA-promoting agents in modulating morphine side effects, the aim of the current study was to address the effects of valproic acid pretreatment on AMT, NAMT, and the possibility of an alteration in the expression of hippocampal *GABAA* receptors underlying morphine tolerance.

## 2. Materials and methods

### 2.1. Ethics statement

Attention was paid to minimize animal suffering during the entire experimental period. Animal studies were designed in accordance to international guidelines and principles. The animal study protocols were approved by local Ethics Committee for Biomedical Research, Tarbiat Modares University, Iran (ID: IR.TMU.REC.1396.563), which is based on the NIH Guide for the Care and Use of Laboratory Animals (NIH publication no 85–23, revised 1996) and Ministry of Health and Medical Education of Iran.

Rats were adapted to the new conditions one week prior to the experiments and were gently handled to avoid unwanted stress associated with handling during housing and experiments. For sacrificing, compressed carbon dioxide gas was used to induce a rapid onset anesthesia. Then rats were rapidly decapitated with a rodent guillotine by an author who was trained and certified for animal handling and humane endpoint application.

### 2.2. Subjects

Male Wistar rats (Pasture institute, Iran) weighing 180–250 g were housed three per cage. Animals were kept in an animal house with stable humidity ($60 \pm 10\%$), temperature ($23 \pm 1°C$), and a 12:12 h light/ dark cycle (lights on at 07:00 am), with free access to food and water, except during experiments. All experiments were carried out between 12:00–15:00.

### 2.3. Drug administration

Fresh morphine solutions were prepared by dissolving morphine sulfate powder (TEMAD, Tehran, Iran) in physiological saline (4 mg/kg) half an hour prior to experiments. Valproic acid (Raha Pharmaceutics, Esfahan, Iran) was administrated one hour prior to morphine injection, and was also freshly prepared by solving valproic acid powder in saline to reach a concentration of 250 mg/kg and were injected subcutaneously.

### 2.4. Experimental design

A total number of 190 rats (including losses) were used in this study (morphine tolerance development, spatial working memory, short-term memory and molecular tests). It should be mentioned that different behavioral and molecular assessments were carried out on same groups of rats by the same experimenter. For evaluating the morphine tolerance development and analgesic response to valproic acid, 186 rats were categorized into the six groups according to the following order: groups 1 and 2 (saline and valproic acid, $n$ = 30 for each group); groups 3 and 4 (non- associative morphine tolerance and non-associative morphine tolerance + valproic acid, $n$ = 36 and 30, respectively); groups 5 and 6 (associative morphine tolerance and associative morphine tolerance+ valproic acid, $n$ = 48 and 12, respectively) (S1 and S2 and 1 and 2 Figs). 60 rats were used for learning and memory analyses ($n$ = 10 rats per group) and 147 rats were used for working memory assessment ($n$ = 30, 30, 30, 30, 12, 15 per groups 1–6,

## Morphine tolerance development

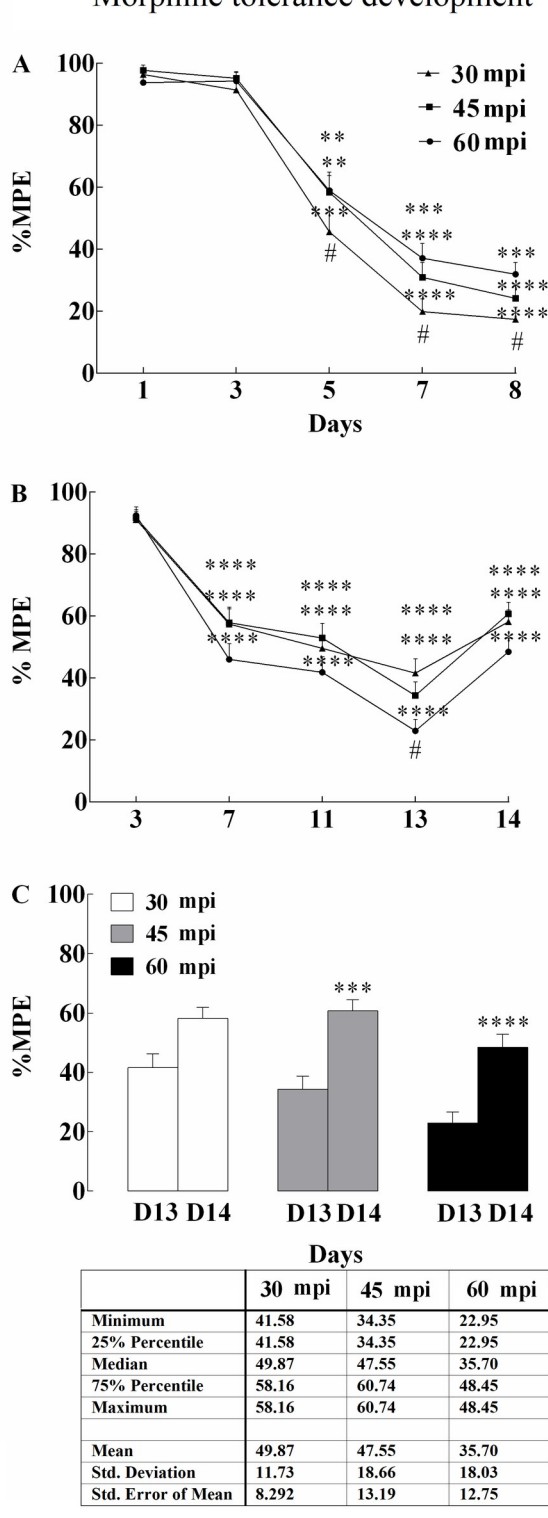

| | 30 mpi | 45 mpi | 60 mpi |
|---|---|---|---|
| Minimum | 41.58 | 34.35 | 22.95 |
| 25% Percentile | 41.58 | 34.35 | 22.95 |
| Median | 49.87 | 47.55 | 35.70 |
| 75% Percentile | 58.16 | 60.74 | 48.45 |
| Maximum | 58.16 | 60.74 | 48.45 |
| | | | |
| Mean | 49.87 | 47.55 | 35.70 |
| Std. Deviation | 11.73 | 18.66 | 18.03 |
| Std. Error of Mean | 8.292 | 13.19 | 12.75 |

**Fig 1. The development of non-associative (NAMT) and associative (AMT) tolerance to the analgesic effects of morphine.** (A) For the development of NAMT, morphine (4 mg/kg) was injected S.C. every day for 8 days and % Maximum Possible Effect (%MPE) was measured at 30, 45, and 60 min after the injection. (B) Development of AMT by administrating morphine (4 mg/kg) on odd-numbered days in the morphine-paired environment and saline on even-numbered days in the saline-paired environment. AMT is a context dependent phenomenon. (C) Tail-flick

latency was measured following morphine injection in the morphine-paired context (D13) and saline-paired context (D14). Animals showed a fully developed morphine tolerance when in the morphine-paired environment (D13), but not when receiving the same dose of morphine in the saline-paired environment (D14) as assessed on 30-, 45, and 60-min post-injection. Data are presented as mean ± SEM, **$P < 0.01$, ***$P < 0.001$, and ****$P < 0.0001$ vs. first injection of morphine. #$P < 0.05$ compares 30 and 60 mpi (two-way repeated measures ANOVA followed by protected Tukey's test for multiple comparisons).

respectively (Figs 3 and 4). Finally, 36 rats were used for mRNA expression and protein level assessments ($n = 6$ in each group).

**2.4.1. Expression of non-associative (NAMT) and associative (AMT) tolerance to the analgesic effects of morphine.** *2.4.1.1. NAMT*. NAMT was assessed using the tail-flick test in the Hargreaves apparatus. All animals received a dose of morphine for 8 days in their home cage. Rodents were kept in their respective partitions of the Hargreaves apparatus for 20 min before being assessed for their tail flick responses. The stimulating beam was set to create a 2–3 s tail flick response before drug administration and the cut off time was set at 10 s to prevent tissue damage. The average of three tests was considered the overall response of each rodent, called baseline latency. Data was expressed as the percentage of maximal possible effect (% MPE). Test latency was assessed 30, 45, and 60 min after morphine administration. %MPE was calculated as follows: (R2-R1)/(COT-R2), where R1 is baseline latency, R2 test latency, and COT is cut off time [17].

*2.4.1.2. AMT*. To examine AMT, animals were exposed to two distinct environments on alternating days. The environments were different in shape, size, color, sound, and brightness during injections, in addition to touch stimulus. On the first and second days, animals were habituated to the tail-flick boxes for 15 min, after which baseline, latency was assessed in the same manner described for NAMT. Rats were transferred to their experimental or test chambers and were administered with physiological saline S.C. after they had habituated to the new context for 10 min. After 20 min, the subjects were then removed from the paired-environment and transferred to the Hargreaves apparatus, where test latency was measured every 15 min for a period of 60 min. On the alternate days (days 3 to 13), animals were given (4 mg/kg) morphine S.C. in their experimental environment, or just saline in their test environment. Following S.C. administrations, animals were given a 20 min habituation phase, after which tail-flick latency was measured once again.

On experimental day (day 14), tail-flick latency was measured for 15 min. Instead of saline, morphine was injected in the saline-paired context to confirm that the tolerance that had developed was associative. Animals were allowed to explore the experimental environment for 20 min before tail flick assessment [6].

**2.4.2. Spatial working memory (SWM) and short-term memory (STM) assessment.** On the day after morphine tolerance assessment had been completed, short-term memory was assessed using the Y-maze apparatus. The Y-maze is a hippocampal dependent task for spatial memory assessment, based on the natural tendency of rodents to alternate in a non-reinforced manner between successive arms to explore previously unvisited areas. The apparatus is basically three similar arms (length 31.4 cm, height 31.4 cm, width 16 cm) interconnected at a central junction at 120° each. Distal extra-maze spatial cues were placed to help rodents find their direction. This assessment consisted of two phases. In the learning phase, one arm was closed, randomly, and rats were allowed to freely explore the other two arms for 10 min. Rodents were left to rest for an hour before commencing the test phase, where rats were reintroduced to the apparatus from the same starting arm, but this time with all arms open. To assess short-term memory, preference to the novel arm within the first 2 min of the test phase was measured by %Entrance, %Time spent, %Subjects that first visited the novel arm [18].

## The effect of VPA on morphine tolerance development

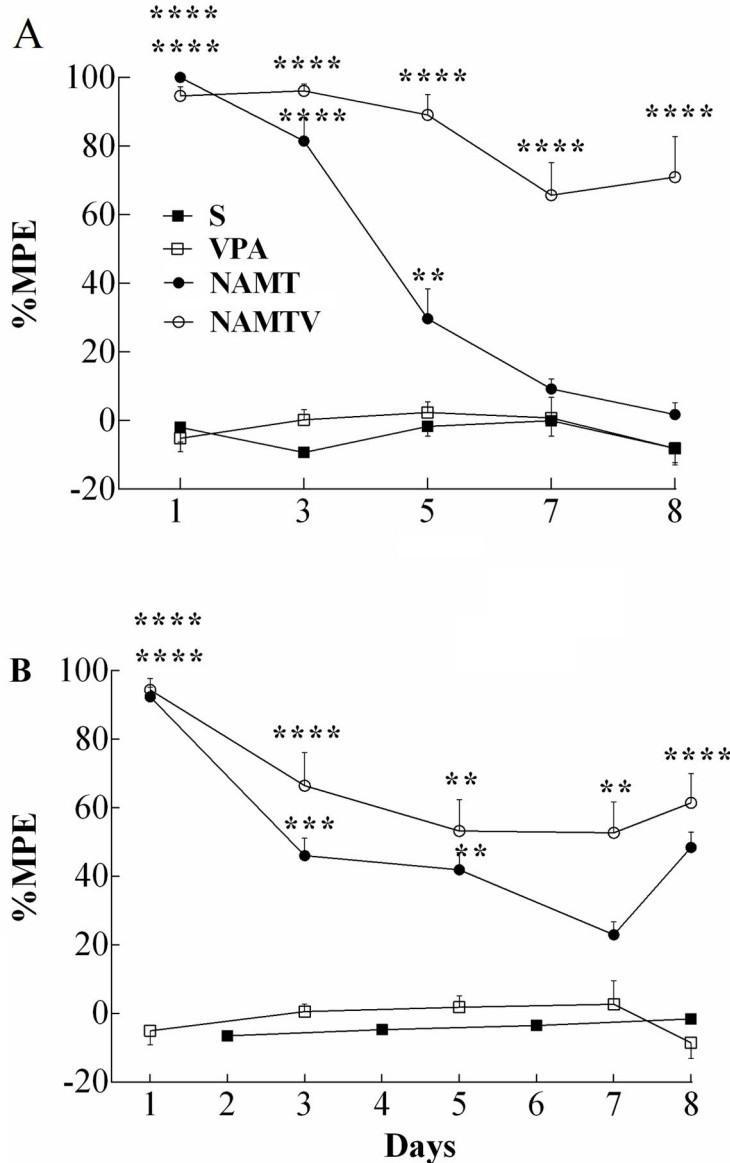

**Fig 2. Valproic acid inhibits analgesic tolerance to morphine. Valproic acid was administered 60 min prior to morphine injection.** (A-B) Tolerance to morphine was developed by repeated injections, while pretreatment with valproic acid one hour before each morphine injection inhibited the expression of tolerance to morphine. Data are expressed as mean ± S.E.M, $^{**}p < 0.01$, $^{***}p < 0.001$, $^{****}p < 0.0001$ as compared to saline (two-way repeated measures ANOVA followed by protected Tukey's test for multiple comparisons). Maximum Possible Effect (MPE), saline (S), valproic acid (VPA), associative morphine tolerance with valproic acid (AMTV), and non-associative morphine tolerance with valproic acid (NAMTV).

After 5–7 days, spatial working memory was assessed using Y-Maze. This assessment consisted of a single 5 min free exploration of the maze, with all arms open. Spatial working

The effects of morphine tolerance + VPA on STM

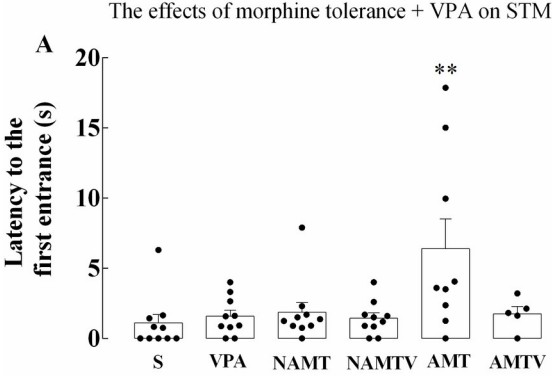

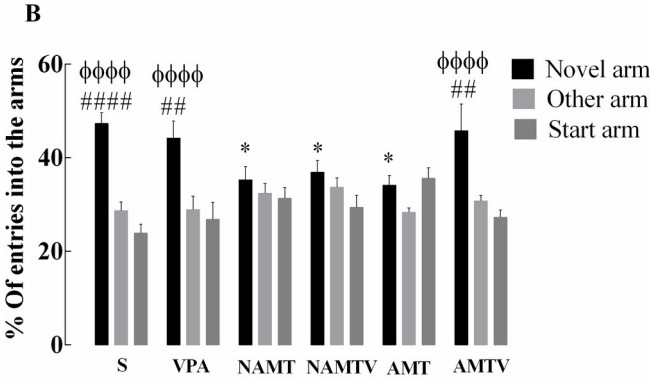

| | Novel arm | Other arm | Initial arm |
|---|---|---|---|
| Minimum | 34.17 | 28.38 | 23.94 |
| 25% Percentile | 35.03 | 28.62 | 26.13 |
| Median | 40.57 | 29.83 | 28.34 |
| 75% Percentile | 46.16 | 32.75 | 32.41 |
| Maximum | 47.35 | 33.69 | 35.64 |
| | | | |
| Mean | 40.62 | 30.48 | 29.08 |
| Std. Deviation | 5.799 | 2.203 | 4.068 |
| Std. Error of Mean | 2.368 | 0.8995 | 1.661 |

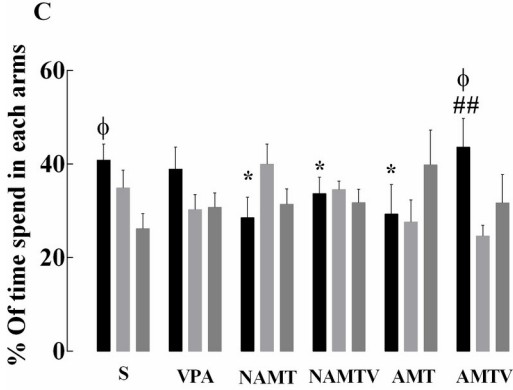

| | Novel arm | Other arm | Initial arm |
|---|---|---|---|
| Minimum | 28.55 | 24.64 | 26.23 |
| 25% Percentile | 29.15 | 26.90 | 29.64 |
| Median | 36.32 | 32.41 | 31.58 |
| 75% Percentile | 41.56 | 36.18 | 33.78 |
| Maximum | 43.63 | 40.03 | 39.83 |
| | | | |
| Mean | 35.84 | 32.01 | 31.96 |
| Std. Deviation | 6.254 | 5.576 | 4.395 |
| Std. Error of Mean | 2.553 | 2.277 | 1.794 |

**Fig 3. Effects of the expression of morphine tolerance and valproic acid pre-treatment on short-term memory, assessed using the Y-maze apparatus with a 1 h ITI.** (A) Latency to first entrance shows a significant enhancement in AMT group. (B) Data shows %Entries into the arms after a 1 h ITI. (C) Shows %time spent in each arm after 1 h ITI. Morphine (4 mg/kg) impaired Y-maze performance. Control and valproic acid groups showed preference to the novel arm, as assessed by %Entries to novel arm. %Time spent in the novel arm decreased in tolerant rats, but this was reversed in AMT after valproic acid pre-treatment. Data has been expressed as mean ± S.E.M, ##P<0.01, ####P<0.0001, ϕ P<0.05, and ϕϕϕ P<0.0001 compares the presence of subjects in the novel arm vs. the other arm and start arm, respectively (two-way ANOVA followed by Tukey's multiple comparison test). *P<0.05 compares novel arm exploration in each group with the control group (one-way ANOVA followed by Holm-Sidak's multiple comparisons test). short-term memory, S, VPA, NAMT, NAMTV, AMT, and AMTV stand for short term memory, saline, valproic acid, non-associative morphine tolerance, non-associative morphine tolerance + valproic acid, associative morphine tolerance, and associative morphine tolerance + valproic acid, respectively.

memory was evaluated by calculating %Spontaneous alternation:

SAP (%) = [(number of correct alternations)/(total arm entries − 2)] × 100. In addition, the total number of arm entries was used as a parameter for locomotor activity.

Animals were injected in the same manner as the tail-flick tests. Drug administration did not stop during the days between spatial working memory and short-term memory assessment to remove the withdrawal effects. Ethanol (70%) was used to clean the Y-maze apparatus between each trial and each subject. The entire data processing and analyses were carried out by blind agents [19].

**2.4.3. Tissue collection.** All the animals were anaesthetized and sacrificed after last behavioral experiment. Hippocampi were immediately dissected and rinsed in ice-cold PBS, snap-frozen in liquid nitrogen and stored at -80˚C until quantitative real time-PCR (qRT-PCR) and western blotting assessments.

**2.4.4. Quantitative real time-PCR experiment.** For gene expression study, total RNA was extracted using Trizol (Invitrogen, Carlsbad, CA, USA) according to the manufacturer's protocol. RNA quality and quantity were determined with agarose gel electrophoresis and nanodrop spectrophotometry via NanoPhotometer NP80 (IMPLEN, Germany). For detection of *GABArα1*, *α2*, *α5* mRNAs, the first-strand cDNAs were synthetized using Easy cDNA reverse transcription kit (Pars tous Biotechnology, Mashhad, Razavi Khorasan Province, Iran) and 2X Real-Time PCR master mix for 40 cycles in a Rotor-Gene Q-Qiagen equipment. The relative mRNA expression levels of *GABArα1*, *α2*, *α5* to GAPDH, as an internal control, were calculated by the $2^{-\Delta\Delta Ct}$ method [20]. The *GABArα2*, GAPDH primers and *GABArα1*, *GABArα5* primers were purchased from Applied Biosystems and SinaClon Companies, respectively. Their specifications have been provided in Table 1 [21].

**2.4.5. Western blotting assessment.** Hippocampi were homogenized in a lysis buffer comprising Tris-HCl, SDS, Triton X-100 and protease inhibitor (Roche, Penzberg, Germany). Protein concentration of each sample was detected using nanodrop spectrophotometry and the equal amounts of protein for each run were loaded. Standardized lysates equivalent to 60 μg of protein was loaded in SDS 12.5% poly acrylamide gel electrophoresis and then transferred to a polyvinylidene difluoride (PVDF) membrane (Chemicon Millipore Co. Temecula, USA). The membranes were blocked in 2% skim-milk to reduce nonspecific binding and then incubated with anti-*GABAr* alpha subunit primary antibody (Merck Co. USA, 1:1000 dilutions) overnight. Subsequently, washing with TBS-Tween 80 was done and incubation with HPR-conjugated (horseradish peroxidase-conjugated) secondary antibody was executed (Sigma Aldrich, St. Louis, MO, USA). Immune reactive polypeptides detection of protein was carried out with ECL reagents (Amersham Bioscience, Piscataway, NJ, USA) and after exposure to X-ray films, the visualized bands were analyzed. To detect β-actin as an internal

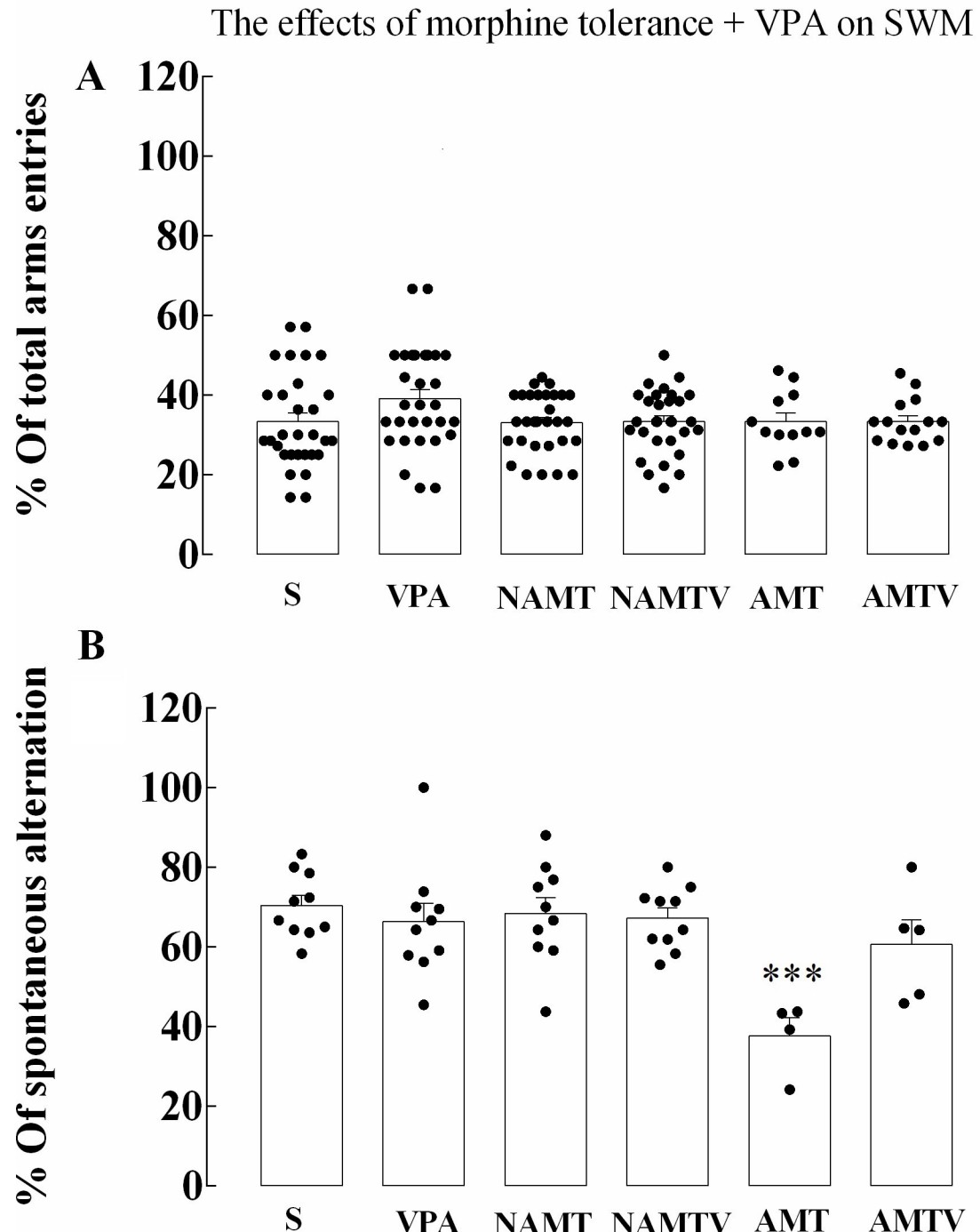

**Fig 4. The effect of the expression of morphine tolerance and valproic acid pre-treatment on spatial working memory.** (A-B) No significant difference was detected in the total number of arm entries, which was used as the index of locomotor activity. However, %Spontaneous alternation in the Y-maze task decreased in AMT. Data has been shown as mean ± SEM, ***p < 0.001 compared with control group (one-way ANOVA followed by Tukey's multiple comparisons test). SWM, S, VPA, NAMT, NAMTV, AMT, and AMTV stand for spatial working memory, saline, valproic acid, non-associative morphine tolerance, non-associative morphine tolerance + valproic acid, associative morphine tolerance, and associative morphine tolerance + valproic acid, respectively.

**Table 1. Sequences of primers used in qRT-PCR.**

| Gene name | | Sequence (5'->3') | NCBI Reference Sequence | TM |
|---|---|---|---|---|
| GABArα1 | Forward | TGCGACCATAGAACCGAAAGA | XM_006246123.4 | 60°C |
| | Reverse | TTGCTGACGCTGTTAAAGGTTTT | | |
| GABArα2 | Forward | GCTTACACGACCTCG | NM_001135779.2 | 60°C |
| | Reverse | GATTCGGGGCGTAGT | | |
| GABArα5 | Forward | GGGAATGGACAATGGAATGC | XM_017589075.2 | 57.5°C |
| | Reverse | TGTCATTGGTCTCGTCTTGTAC | | |
| GAPDH | Forward | AACCCATCACCATCTTCCAG | NM_017008.4 | 59°C |
| | Reverse | CACGACATACTCAGCACCAG | | |

GABArα1; Gamma-aminobutyric acid type A receptor alpha 1 subunit.

GABArα2; Gamma-aminobutyric acid type A receptor alpha 2 subunit.

GABArα5; Gamma-aminobutyric acid type A receptor alpha 5 subunit.

GAPDH, Glyceraldehyde-3-phosphate dehydrogenase.

control, blots were stripped in stripping buffer (pH = 6.7) and then probed with anti β-actin antibody (1:1000, Cell Signaling Technology Co. New York, USA). *GABArα* bands in each group were normalized to their related β-actin bands. Relative intensity of the protein bands was measured by ImageJ software [22].

**2.4.6. Statistical analysis.** The data has been presented as mean ± SEM. The data for morphine tolerance development was analyzed by two-way repeated measures ANOVA, followed by protected Tukey's test for multiple comparisons. Western blot analysis, gene expression assessment and % of spontaneous alternation were analyzed by one-way ANOVA followed by a post-hoc Tukey's test. Memory assessments were analyzed using two-way repeated measures ANOVA followed by protected Tukey's test. All calculations were done with GraphPad PRISM software version 6.0 (GraphPad software Inc., San Diego, CA, USA). P-value <0.05 was considered significant.

# 3. Results

## 3.1. Behavioral experiments

**3.1.1. NAMT and AMT flourish in a time dependent manner.** According our results, the first injection revealed a significant analgesic effect of morphine compared to the control group. Rodents received daily S.C. doses of 4 mg/kg for 8 days during the tolerance assessment experiment. Tolerance to the anti-nociceptive effect of morphine developed from the second injection and was eventually expressed within five days as a result of chronic morphine administration. Our data from 36 rats showed that tolerance peaked at 30 min post injection (mpi) and remained at its peak until 60 mpi (F (1, 204) = 25.67; p<0.0001; S1A–S1C Fig). At 30 mpi, tolerance development accelerated and had the highest rate of change and largest area under curve compared to other assessed mpis. Saline administration had no significant effect on % MPE in the non-associative group (Fig 1A and Table 2).

After the NAMT experiment was completed, a separate group was trained in the associative context, where saline was injected on even-days in the saline-paired environment, and morphine (4 mg/kg S.C.) was administrated on odd-numbered days in the morphine-paired environment. Baseline latency was measured on experimental days 1 and 2, respectively. By the sixth administration (day 13), animals showed tolerance to morphine in the morphine-paired environment (S2A–S2F Fig; n = 48). Interestingly, our data revealed that morphine produced analgesia when rodents were injected with morphine in the saline-paired environment on day

**Table 2. Comparison between area under curve and slope in NAMT and AMT.**

| Type of tolerance | MPI | Slope (Time/MPE) | Area under curve (Time/MPE) |
|---|---|---|---|
| NAMT | 30 | 22.71±5.21 | 53.42±2.96 |
| | 45 | 20.03±5.72 | 61.43±2.94 |
| | 60 | 18.06±7.48 | 63.27±2.98 |
| AMT | 30 | 4.85±1.31 | 109.906±5.36 |
| | 45 | 5.60±1.48 | 101.51±4.14 |
| | 60 | 6.01±1.18 | 89.47±4.83 |

Compression between area under curve and slope in both types of tolerance. Values are expressed as mean ±SEM. AMT = Associative morphine tolerance; NAMT = non-associative morphine tolerance.

14 (F (1, 310) = 45.95; p<0.0001). AMT was measured as a significant difference in %MPE between day 13 and 14. Maximum AMT occurred at 60 mpi. At 60 mpi we observed an accelerated AMT trend until day 14, with the highest rate of change occurring between days 13 and 14 compared to other time intervals (Fig 1B and 1C and Table 2).

**3.1.2. Systemic valproic acid inhibits the expression of AMT and NAMT.** To evaluate the effect of valproic acid on the morphine tolerance expression, rats received systemic administration of valproic acid (250 mg/kg S.C.) 60 min before morphine injection. The tail test was conducted in the same manner explained previously. As shown in Fig 2, valproic acid injections reversed both AMT and NAMT (F (3, 36) = 179.9; p<0.0001; (n = 6–10). In other words, morphine still produced analgesic effects on day 7 when the rodent was pretreated with valproic acid. Although tolerance had developed, nevertheless, our tail flick results show that analgesia remained significantly higher in groups that received valproic acid prior to morphine compared to the control group. In morphine injected rats that did not receive valproic acid pretreatment, %MPE fell to the level of control group on day 7 (Fig 2A and 2B). Our results did not show any significant difference in %MPE between saline and valproic acid groups.

**3.1.3. Systemic valproic acid improves impaired short-term memory due to AMT.** The latency to the first entrance and the time it takes for a subject to enter an arm were measured as short-term memory parameters in Y- maze task. Our behavioral data demonstrated that latency to the first entrance was significantly affected by AMT (F (5, 48) = 4.005; p = 0.0041, n = 5–10). This index shows that the performance of AMT rats was impaired (6. 40%) compared to control (1.09%). The number of subjects that entered the novel arm as their first choice showed no significant difference across the experimental groups (Fig 3A).

The Y-maze apparatus was also used to evaluate the effect of valproic acid pretreatment on short-term memory in the morphine tolerant model. Our results showed that the number of entrances to the novel arm was depressed significantly in both NAMT and AMT groups (F (5, 156) = 7.182; p<0.05). Systemic administration of valproic acid did not change %Entrance to the novel arm in comparison to control rats (Fig 3B). A similar pattern was observed in %time spent in the novel arm. Morphine tolerance led to a significant decrease in the time that animals spent in the novel arm compared to control. In addition, a significant increase was seen in time spent in the novel arm as a result of valproic acid administration in AMT when compared with control animals (Fig 3C).

**3.1.4. Spatial working memory is disrupted in AMT.** As showed in Fig 4, one-way ANOVA was used to determine whether the expression of morphine tolerance affected working memory. To assess locomotion and spatial working memory performance, % of total arm entrances and % of spontaneous alternation were measured through Y-maze. All animals demonstrated an intact performance in the Y-maze, showing no significant difference in

locomotor activities either (F (5, 141) = 1.740; p = 0.1294). Our data showed that spatial working memory was not affected by valproic acid administration or morphine tolerance, although AMT resulted in reduced %SPA.

### 3.2. Molecular assessments

**3.2.1. Systemic valproic acid decreases the expression of *GABArα* subunits in morphine tolerant rats.** The expression level of *GABArα* subunits mRNAs were assayed in the hippocampus region of all groups. Regarding to Fig 5, the results showed that valproic acid can reduce the expression of *GABArα1*, *α2* and *α5* subunits to 20%, 17% and 15% of control level, respectively (F (5, 30) = 25.36; P < 0.0001). In NAMT group, morphine did not alter the expression of *GABArα2* and *α5* significantly; while increased *α1* subunit by fold about 1.86 related to control level (p < 0.05). Pretreatment with valproic acid, 30 min before morphine administration reduced the gene expression of *GABArα1*, *α2* and *α5* subunits by about 53%, 27% and 40% in compare with control group, respectively. In contrast, the expression of *GABArα1*, *α2* and *α5* subunits in AMT group were significantly enhanced about 2.16, 2.33 and 2.5-fold. Furthermore, after valproic acid pretreatment there is a general trend for enhancement when compared with control samples (p < 0. 01).

**3.2.2. The increased *GABArα* protein level in AMT group is not altered by systemic valproic acid treatment.** To determine the effect of valproic acid on *GABArα* protein, we assessed the levels of alpha subunit by western blot analysis in tolerate rats F (5, 30) = 25.36; P < 0.0001). The alpha subunit level significantly decreased about 80% of control in valproic acid group as shown in Fig 6 (*p* < 0.05). In NAMT and AMTV groups the expression of alpha subunit increased, although only in AMT group changed significantly (*p* < 0.0001). These fold changes in alpha protein level were 1.83 and 2.5 relative to control group, respectively. On the other hand, valproic acid pretreatment in NAMTV rats caused a significant decline about 0.67 in comparison to saline group. The level of alpha subunit slightly decreased in AMTV group but it was not statistically significant.

## 4. Discussion

Herein AMT and NAMT expression were suppressed by valproic acid pretreatment, as observed in both the behavioral and molecular assays. Several behavioral studies have focused on NAMT, while physiological experiments have rarely assessed the procedures that can help to investigate the effects of AMT expression directly. Although tolerance was expressed in both AMT and NAMT groups following repeated morphine injections, AMT rats were significantly more morphine-tolerant than NAMT ones, as measured by %MPE.

In the first step, we evaluated the behavioral performance of rats in different groups. It seems that the induction of these two types of tolerances follow different mechanisms. AMT was not restricted to cellular adaptations of a specific receptor or second messenger cascade like those that influenced NAMT [23]. AMT is based on integrated associative learning factors and can be justified based on Pavlovian classical conditioning rules [24, 25]. Baker and Tiffany assumed that an endogenous compensatory mechanism designed to maintain homeostasis may play a crucial role in drug effects or stimuli associated with a drug (24). Furthermore, Grisel et al. in 1996 argued that—unlike NAMT—the mechanism underlying AMT is not mediated by NMDA receptors. In addition, selective antagonists of dopaminergic system did abolish AMT, but not NAMT [25].

Since GABAergic inhibitory currents are involved in tolerance, valproic acid pretreatment, which is a GABA-enhancer, may lead to attenuation of morphine tolerance. It should be noted

The expression of GABA-A subunits in the hippocampus

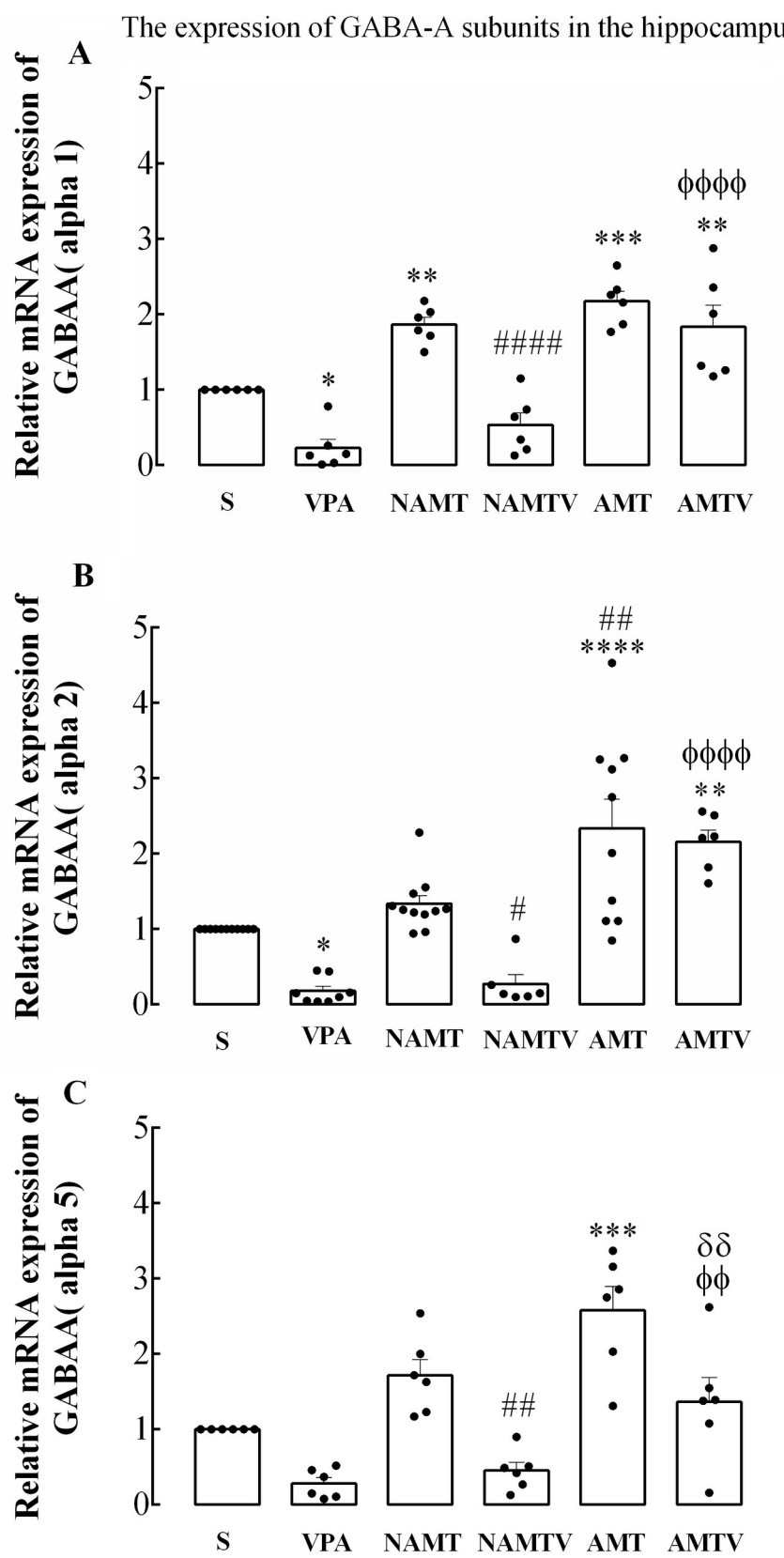

**Fig 5. Quantification data for qRT-PCR analysis of changes in the expression of *GABArα1*, *α2* and *α5* subunits in the hippocampus of rats.** Graph showing treated and control values in different experimental groups and GAPHD was used as an internal control. Data are shown as mean ± SEM, *p < 0.05, **p < 0.01, ***p < 0.001, ****p < 0.0001 compared with control group. #, φ, σ indicate a significant difference compared with NAMT, NAMTV and AMT, respectively (one-way ANOVA followed by Tukey's multiple comparisons test). S, VPA, NAMT, NAMTV, AMT and AMTV mean, saline, valproic acid, non-associative morphine tolerance, non-associative morphine tolerance+ valproic acid, associative morphine tolerance and associative morphine tolerance+ valproic acid, respectively.

that our data also showed that valproic acid did not cause analgesic effects on its own, which is consistent with previous reports [26, 27].

It has been demonstrated that valproic acid inhibits GABA transaminase, glutamic acid decarboxylase, histone deacetylase, and SSADH enzymes while enhancing the PENK system in the rat brain [28, 29]. Valproic acid is a multi-effect drug, which increases inhibitory currents of *GABA* in bipolar disorder [30], morphine dependency [31], epilepsy [32] and neuropathic pain [33]. In accordance with these studies, our results demonstrated that valproic acid suppresses tolerance to the analgesic effects of morphine.

In the present study, we attempted to investigate the effect of both types of tolerance on spatial working memory and short-term memory, and how valproic acid affects them. Short-term memory refers to the capacity of the mind to temporarily maintain information over a period

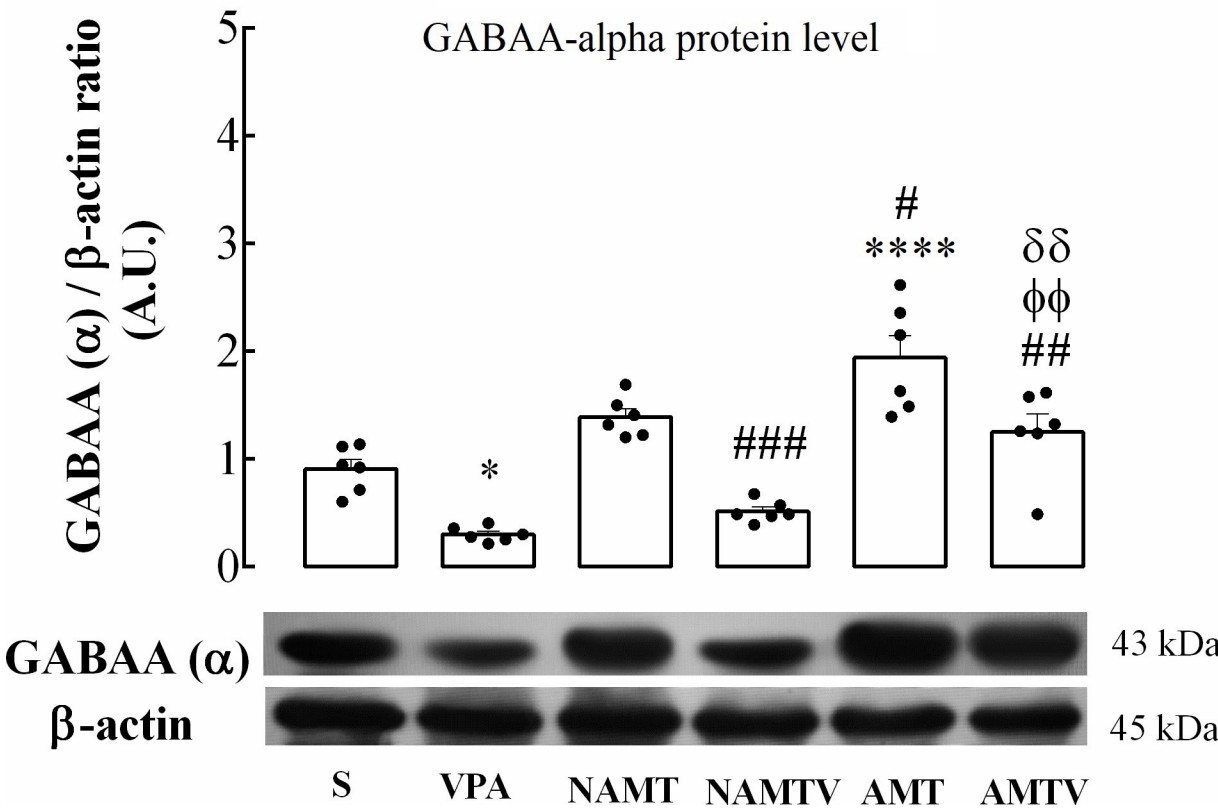

**Fig 6. The effects of morphine tolerance and valproic acid pretreatment on hippocampal *GABArα* protein density was separated on SDS–PAGE, western blotted, probed with specific primary antibody, and reported with anti β-actin antibody.** Data are shown as mean ± SEM, *p < 0.05, **p < 0.01, ***p < 0.001, ****p < 0.0001 compared with control group. #, φ, σ indicate a significant difference compared with NAMT, NAMTV and AMT, respectively (one-way ANOVA followed by Tukey's multiple comparisons test). A.U., S, VPA, NAMT, NAMTV, AMT and AMTV mean arbitrary unit, saline, valproic acid, non-associative morphine tolerance, non-associative morphine tolerance + valproic acid, associative morphine tolerance and associative morphine tolerance+ valproic acid, respectively.

of seconds without manipulating it. Although short-term memory is considered to be part of spatial working memory, these two terms have different concepts. Spatial working memory is considered a workplace for structures and processes used for controlled manipulation of limited information before recall. Spatial working memory plays a crucial role in executing a wide span of cognitive functions [18]. One of the explanations proposed about how spatial working memory works is that it includes several subcomponents, such as the central executive function, visuospatial information, phonological loop, and episodic buffer. Only two of these theoretical frameworks are related to short-term memory mechanisms—the phonological loop and the visuospatial sketchpad [34].

We observed no significant differences across the groups in the locomotion. However, the absolute and percentage of latency to first entrance and spontaneous alteration were unaffected by NAMT development, while we saw a significantly enhanced decision index and attenuated spontaneous alternation in AMT.

Based on our short-term memory data, a significant impairment of memory was seen in both types of tolerance development during chronic morphine exposure in comparison with control group. In line with our results, several investigations showed that morphine injection leads to impaired working memory, spontaneous alteration, and disruption in reference memory acquisition in radial and Y-Maze via various hippocampal tissue alterations such as enhancement of lipid peroxidation and inflammation [35, 36]. In contrast with our findings related to locomotor activity, acute morphine injection resulted in hyper-activity in the Y-maze after a 1 h ITI [37]. Also, our data indicated that spatial working memory impairment induced by opiate administration was related to the dose and administration method, which might explain the significant reduction we observed in spontaneous alternation of the AMT group. It has been shown that following acute injection of morphine, an impaired efficacy in spatial working memory components occur in the Y-maze and 8-arms radial maze tasks [36], although not in the Morris water maze [38]. Therefore, morphine has complicated effects on spatial working memory and short-term memory, which can be influenced by the type of task, gender of subjects, and route of drug administration.

It seems that in associative tolerance, the hippocampus is a locus for the consolidation and acquisition phase of memory. Mitchell and colleagues showed that the projections from CA1 to the L/BL amygdala were involved in associative tolerance during morphine injection paired to a distinct context [6]. One of the possible mechanisms assumed to be involved in impaired memory retrieval in the Y-maze may be related to a manifest attenuation in synaptic performance during morphine exposure and AMT development. Chronic morphine exposure as an addictive drug may abuse brain mechanisms of hippocampal LTP behind spatial learning and memory and elicits a specific synaptic potentiation, which is the same as LTP in the brain [39]. Contrarily, it is intriguing that hippocampal neurons which are participated in learning and memory formation overlap with which are involved in processing and assigning value to drug-associated cues in tolerance [40]. In associative tolerance, the required spatial relationship between environmental cues and drug administration has been established, induce several compensatory functions, long term metaplastic changes and aberrant memory formation occurred in this model [41]. The CA1 has a critical role in match–mismatch discrimination that occurs in morphine-associated context exposure along the acquisition phase of associative tolerance, Therefore, associative tolerance will be trigger an aberrant LTP in hippocampal connections [6].This potentiation reduces the capacity of hippocampal synapses to develop new plasticity and leads to learning and memory disruption in AMT rats [42]. Furthermore, AMT uses additional learning circulates related to several parts of the brain to develop its associative aspects. It seems that the cumulative effects of synaptic saturation led to additional memory impairment in AMT.

Previous studies documented an impairment of performance in water maze and learning ability deficiency in 8-arms maze due to the inhibition of brain cholinergic system by morphine tolerance [43–45]. Furthermore, the amygdala is involved in the encoding of affective reward-related memory associated with particular environmental cues during morphine exposure [46]. Amygdala neurons modify their firing pattern in response to associative aspects of morphine tolerance [47]. It was demonstrated that some opioid receptor agonist—acting as a memory destructive—is essential to associative, but not non-associative, morphine tolerance [6]. It has been reported that inhibiting tonic GABA discharge through morphine tolerance resulted in enhanced serotonin and epinephrine levels into the amygdala and hippocampus. The increased levels of serotonin caused to hypersensitivity and memory impairment in rodent and valproic acid was able to decrease the morphine tolerance and ameliorate the memory impairment through enhanced GABAergic currents [48].

On the other hand, valproic acid treatment decreased memory deficits, increased reaction time, and improved cognitive impairment in Alzheimer's disease [49], CSE model of rats, [50] and seizure development [51, 52]. Considering these studies, we suspected that an enhancement of novel arm exploration, indexed by %total time spent in the novel arm and the number of entrances to the novel arm, occurred after valproic acid pretreatment, but it was not potent enough to prevent the impairment of learning.

In the next step, molecular aspect of our study was on the effect of morphine tolerance and valproic acid-morphine co-administration on *GABArα1*, *α2* and *α5* mRNAs expression. Quantitative real-time PCR revealed that valproic acid pretreatment decreased *GABArα* mRNA expression only in the NAMTV group.

Several reports indicating that the *GABArα1* is the major subtype, involved in about 50% of all *GABArs* in the brain. Previous documents were reviewed that an enhanced anxiety behavior including less time spent in the open field's center and increased auditory startle responses was observed in *GABArα1* mutant rats [53]. Indeed, other evidences demonstrated that exposure to cocaine and morphine stressors reduced *GABArα1* expression in the mPFC (Medial prefrontal cortex) and resulted cognitive and design making impairments [54].

It was noted that impairment of memory retention is related to *GABArα2* activation, and reduction in GABAergic currents increases the possibility of seizure occurrence and early mortality in *GABArα2* mutant mice [55]. Furthermore, memory retention is reduced in the presence of the *GABAA* receptor agonist [56]. The Alpha2 subunit of *GABAA* receptors, which we measured in this study, is encoded by *GABArα2* and expressed vastly in reward circuits such as hippocampus and dopaminergic neurons. This subunit is involved in addiction, specifically during adolescences, and elevates the risk of addiction during poor neural connectivity in reward-loop [57, 58]. Furthermore, it was estimated that *GABArα2* affected the mean strength of the reward network, decision making, and impulsivity in healthy individuals, while this pattern was opposite in heroin users due to the long lasting effects of repeated drug exposer [59]. the *GABArα5* is mainly expressed in layer 5 of pyramidal cells and associated selectively with cognitive function. Available evidence indicates that the hippocampal dependent performance in fear conditioning, appetitive conditioning [60], Morris water maze [61] and novel object recognition were improved in *GABArα5* point-mutated mice [62]. Further these knowledges, it was shown that a deficit in short-term memory in a particular puzzle box and tasks characterized by high memory interference included behavioral pattern separation was apparent in mice with *GABArα5* modified expression [63]. According to memory- ameliorative findings, dugs acting at the benzodiazepine site of *GABArα5* are candidate to enhance performance in learning and memory [64].

Finally, our study on morphine tolerance assumed that the widespread increase in *GABArα* receptor protein in AMT group may be due to synaptic saturation during strong connectivity of the reward and learning networks. Moreover, the mentioned increment was not influenced by valproic acid pretreatment.

## 5. Conclusion

The present study tested the effects of the anticonvulsant, mood-stabilizing, and *GABA* facilitator, valproic acid on the behavioral tolerance induced by morphine. It was figured out that both learned and non-associative morphine tolerance influence short-term memory and the subjacent expression of *GABArα* subunits. The findings add to our understanding of the behavioral and molecular aspects of the learned tolerance to morphine effects, but further studies are required to clarify the mechanisms responsible for these changes in memory and *GABArα* subunits expression.

## Supporting information

**S1 Fig. Analgesic effect of morphine (4 mg/kg) was decreased after consecutive daily injections in NAMT.** Changes in tail-flick responses were expressed as percentage of maximal possible effect in three time-courses (30, 45, and 60 min past morphine injection. A, B and C, respectively). Values were expressed as mean ± S.E.M, ($^{*}$P $<$ 0.05, $^{**}$P $<$ 0.01, $^{***}$P $<$ 0.001 and $^{****}$P $<$ 0.0001 vs. control (saline injection) group. 1, 3, 5, 7 and 8 represented days post subcutaneous injections (Repeats-measured two-way ANOVA followed by protected Tukey's test for multiple comparisons), "MPE and NAMT "mean maximal possible effect and non-associative morphine tolerance, respectively.
(TIF)

**S2 Fig. AMT flourished in a time dependent manner (30, 45, and 60 related to A, B and C respectively).** Acquisition of AMT was induced by administration of saline on even days and morphine (4 mg/kg) on odd-days in their distinctive context until day 13. On day 14 morphine was injected in saline-paired environment and %MPE was measured at three time-courses (30, 45, and 60). (D-E-F) AMT is dependent to the injection context. % MPE in morphine-paired and saline-paired context before (baseline) and after (morphine in day 13 and 14) the development of AMT. baseline's MPE was measured on experimental days 1 and 2, respectively. Animals were tolerant to morphine when tested in the morphine-paired environment on day 13th but day 14th, anti-nociceptive effects of morphine were significantly increased with the same dose of morphine given in the saline-paired environment. Data were expressed as mean±S.E. M, ($^{*}$P $<$ 0.05, $^{***}$P $<$ 0.001 and $^{****}$P $<$ 0.0001 vs. control (saline injection) group and baseline. 1–14 represented days post injection (Repeats-measured two-way ANOVA followed by protected Tukey's test for multiple comparisons). "MPE and AMT" mean maximal possible effect and associative morphine tolerance, respectively.
(TIF)

**S1 Data. The dataset of non-associative morphine tolerance (NAMT) development.**
(XLSX)

**S2 Data. The dataset of associative morphine tolerance (AMT) development.**
(XLSX)

**S1 File. The dataset of morphine analgesic effect (4 mg/kg) after consecutive daily injections in non-associative morphine tolerance.** Experiment 1.
(DOCX)

**S2 File. The dataset of morphine analgesic effect (4 mg/kg) after consecutive daily injections in associative morphine tolerance.** Experiment 2.
(DOCX)

**S3 File. The dataset of morphine tolerance expression and VPA pre-treatment on STM, assessed using the Y-maze apparatus with a 1 h ITI.** Experiment 3.
(DOCX)

**S4 File. The dataset of morphine tolerance expression and VPA pre-treatment on SWM.** Experiment 4.
(DOCX)

**S5 File. Quantification dataset for qRT-PCR analysis of changes in the expression of** *Gabrα1*, *α2* **and** *α5* **subunits in the hippocampus of rats.** Experiment 5.
(DOCX)

**S6 File. The dataset of morphine tolerance and VPA pretreatment on hippocampal** *Gabrα***, probed with specific primary antibody, and reported with anti β-actin antibody.** Experiment 6.
(DOCX)

**S1 Raw images. The image of chemiluminescence western blot probed with** *Gabrα* **and re-probed by β-actin anti-bodies.**
(PDF)

## Author Contributions

**Conceptualization:** Yaghoub Fathollahi.

**Data curation:** Yaghoub Fathollahi.

**Formal analysis:** Yaghoub Fathollahi.

**Funding acquisition:** Yaghoub Fathollahi.

**Investigation:** Ghazaleh Ghamkharinejad, Seyed Hossein Marashi.

**Methodology:** Ghazaleh Ghamkharinejad, Seyed Hossein Marashi, Forough Foolad, Mohammad Javan.

**Project administration:** Yaghoub Fathollahi.

**Resources:** Yaghoub Fathollahi.

**Software:** Seyed Hossein Marashi, Yaghoub Fathollahi.

**Supervision:** Yaghoub Fathollahi.

**Validation:** Yaghoub Fathollahi.

**Writing – original draft:** Ghazaleh Ghamkharinejad, Seyed Hossein Marashi.

**Writing – review & editing:** Ghazaleh Ghamkharinejad, Seyed Hossein Marashi, Forough Foolad, Mohammad Javan, Yaghoub Fathollahi.

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
