## [Decision Letter · Decision Letter 0]

7 Dec 2020

PONE-D-20-29685

Systemic sodium valproate ameliorates impaired short-term memory induced by associative morphine tolerance in rats

PLOS ONE

Dear Dr. Fathollahi,

Thank you for submitting your manuscript to PLOS ONE. After careful consideration, we feel that it has merit but does not fully meet PLOS ONE’s publication criteria as it currently stands. Therefore, we invite you to submit a revised version of the manuscript that addresses the points raised during the review process.

In addition to the the reviewer's comment (attached), I am also of the opinion that the manuscript needs major editing of the language and also demonstrate changes in GABAA α2  protein in the hippocampal slices by immunohistochemical analysis, since in changes in mRNA levels does not translate  in the expression the respective protein as mRNA undergoes transcriptional modifications. all the changes you require for acceptance versus which changes you recommend. 

The manuscript needs to revised according to the comments of the editor and the reviewer before reconsideration of the manuscript.

We look forward to receiving your revised manuscript.

Kind regards,

Prasun K Datta, Ph.D

Academic Editor

PLOS ONE

Journal Requirements:

2. Thank you for including your ethics statement:  "Tarbiat Modares University".   

a. Please amend your current ethics statement to confirm that your named ethics committee specifically approved this study.

For additional information about PLOS ONE submissions requirements for ethics oversight of animal work, please refer to http://journals.plos.org/plosone/s/submission-guidelines#loc-animal-research  

4. Please include your tables as part of your main manuscript and remove the individual files. Please note that supplementary tables should remain as separate "supporting information" files.

Reviewers' comments:

Reviewer's Responses to Questions

**Comments to the Author**

1. Is the manuscript technically sound, and do the data support the conclusions?

Reviewer #1: Partly

2. Has the statistical analysis been performed appropriately and rigorously? 

Reviewer #1: Yes

3. Have the authors made all data underlying the findings in their manuscript fully available?

Reviewer #1: Yes

4. Is the manuscript presented in an intelligible fashion and written in standard English?

Reviewer #1: No

5. Review Comments to the Author

Reviewer #1: Reviewer Recommendation and Comments

Manuscript PONE-D-20-29685

The present study tested the effects of the anticonvulsant, mood-stabilizing, and GABA facilitator sodium valproate on the behavioral tolerance induced by morphine. The authors first investigated the development of unconditioned (non-associative response) and learned tolerance (context-dependent response) to the analgesic effects of morphine using the tail-flick test. Then they examined morphine tolerance on hippocampal-dependent tasks named spatial-working and short-term memory procedures using the Y-maze apparatus. Finally, the authors examined the changes in the expression of hippocampal GABAA receptors underlying morphine tolerance using the quantitative real-time PCR technique to detect GABAA �2 mRNA. Overall, the authors report that both learned and non-associative morphine tolerance influence short-term memory and the subjacent expression of GABAA �2 mRNA. Overall, the results are interesting and add to our understanding of the behavioral and molecular aspects of the learned tolerance to morphine effects. Morphine is used as a painkiller to treat moderate, severe, and chronic pain. Considering that those suffering from chronic pain have the potential to misuse their medication and taking into account that morphine has the potential to be highly addictive, as tolerance to it develops rapidly, any additional information that can be applied to the potential abuse liability is extremely relevant and timely. However, before the manuscript can be considered for publication, the authors should address several points. Specific concerns are provided below:

- In General, the manuscript is understandable but it will benefit from a grammar correction.

- Page numbers are missing.

- First page of the Introduction, line 7: MOR (what does it mean?)

- Methods section (Drugs administration): The authors must inform the basis for the choice of drug dose and drug dosages.

- If I understand, regarding the learned tolerance (section 2.3.1.2), it seems that the tail-flick analysis was performed out of the morphine-related context. The authors are invited to explain in more detail how learned morphine tolerance can be evoked when the test of analgesia is performed in the absence of contextual cues previously associated with drug effects.

- Statistical analysis is correctly informed but, in the Results section, the data related to the statistical inferences resulting from ANOVA are missing (degrees of freedom, F values, probability).

- There are some problems related to the number of figures and figures described in the manuscript. For instance, in section 3.1, the last paragraph, the authors inform: … (Fig 1 B-C). However, C is not a part of figure 1. Indeed, graphic C is a part of the supplementary material. Moreover, besides the poor quality, the figures are difficult to understand and the small fonts used do not help the reader. However, the biggest problem is the excessive use of asterisks (mainly in figures 3 and 4) that turns the figures absolutely incomprehensible to me.

- Section 3.4: the authors defined what they call “decision-making criteria”, or “decision index”. Have other studies used this same approach to name the first latency to enter an arm of a maze?

- The discussion of the data is difficult to follow without a good analysis of the figures provided. Therefore, before the manuscript can be considered for publication, the authors should address the concerns provided above.

6. PLOS authors have the option to publish the peer review history of their article (what does this mean?). If published, this will include your full peer review and any attached files.

Reviewer #1: No

---

## [Author Response · Author response to Decision Letter 0]

13 Apr 2021

Comments to the Author

 Specific concerns are provided below:

- In addition to the reviewer's comment (attached), I am also of the opinion that the manuscript needs major editing of the language and also demonstrate changes in GABAA α2 protein in the hippocampal slices by immunohistochemical analysis, since in changes in mRNA levels does not translate in the expression the respective protein as mRNA undergoes transcriptional modifications.

According to reviewer's comments the manuscript needs to explant the changes in GABAA α2 protein at the hippocampus region by immunohistochemical analysis.

Since there was no pre-fixed tissue to cut for IHC, and due to restrictions imposed by the covid-19 pandemic as well as ethical issues to include a new animal cohort, we used Western blotting to confirm changes in protein level. Fortunately the frozen hippocampal tissue of all experimental groups were available. 

Antibody against GABAA-alpha subunit was used for Western blotting. We also included mRNA expression data for all GABAA alpha 1, 2 and 5. The new results are presented in Fissures 5 and 6. 

1- In General, the manuscript is understandable but it will benefit from a grammar correction.

It was modified in the manuscript and send as ‘Revised Manuscript with Track Changes’.

2- Page numbers are missing.

It was added to manuscript) p1-32)

3- First page of the Introduction, line 7: MOR (what does it mean?)

It was referred to Mu-opioid receptors (p4p1l5)

4- Methods section (Drugs administration): The authors must inform the basis for the choice of drug dose and drug dosages.

The authors have exerted every effort to ensure that drug dosage selection in this manuscript was in accord with current recommendations in literatures and the methods were applied in pervious experiments and published paper from our laboratory for S.C. morphine administration to induce tolerance in 8 days. We chose to administer morphine S.C. because of the simplicity and low-pain that it induces, and we get maximal effecting dose fast. 

However, in view of our ongoing research in new tolerance induction methods (AMT), the environmental cues effects on drug reactions in brain and for checking the new package of morphine and inserted information for each new employed drug, the basic pilot experiments (2.5, 3, 4. 5 and 10 mg/kg morphine sulfate ) were applied to determine the effective dosage of morphine and VPA to get maximal analgesic effects according to % of MPE calculated in Hargraves apparatus. 

5- If I understand, regarding the learned tolerance (section 2.3.1.2), it seems that the tail-flick analysis was performed out of the morphine-related context. The authors are invited to explain in more detail how learned morphine tolerance can be evoked when the test of analgesia is performed in the absence of contextual cues previously associated with drug effects.

Although, the tail-flick test was performed out of the context associated with morphine administration, the injections of morphine were done in context. The effects of environmental cues are more essential during the morphine injection. 

On the basis of a Pavlovian conditioning analysis of morphine tolerance, cues signaling the systemic effects of morphine come to elicit conditional pharmacological responses that attenuate the effect of the drug. The context-induced tolerance can be useful tool for studying the mechanisms of short term and long-lasting central effects of morphine. According to literatures the stimulating effect of morphine-conditioned context appeared immediately after drug injection and could be calculate even the animal was moved to another room for testing immediately, then %MPE were calculated after transfer animal to Hargraves apparatus without eliminating environmental cues effects.

- Statistical analysis is correctly informed but, in the Results section, the data related to the statistical inferences resulting from ANOVA are missing (degrees of freedom, F values, probability).

The edited p-values according to editor’s comments are inserted in manuscript:

P11p2l6, p11p3l8, P13p1l4, P14p1l1, p14p2l3, p15p2l5, p16p1l3, p17p1l2

- There are some problems related to the number of figures and figures described in the manuscript. For instance, in section 3.1, the last paragraph, the authors inform: … (Fig 1 B-C). However, C is not a part of figure 1. Indeed, graphic C is a part of the supplementary material. Moreover, besides the poor quality, the figures are difficult to understand and the small fonts used do not help the reader. However, the biggest problem is the excessive use of asterisks (mainly in figures 3 and 4) that turns the figures absolutely incomprehensible to me.

Thank you for the comment, it was edited in revised manuscript. 

#

* indicated significant deference between novel arm exploration in each group with the control group

# indicated significant deference between novel arm and other arm

ϕ indicated significant deference between novel arm start arm

- Section 3.4: the authors defined what they call “decision-making criteria”, or “decision index”. Have other studies used this same approach to name the first latency to enter an arm of a maze?

It was modified in the manuscript p15p2l1-7

- The discussion of the data is difficult to follow without a good analysis of the figures provided. Therefore, before the manuscript can be considered for publication, the authors should address the concerns provided above.

it was modified in the manuscript

---

## [Decision Letter · Decision Letter 1]

26 May 2021

PONE-D-20-29685R1

Unconditioned and learned morphine tolerance influence hippocampal-dependent short-term memory and the subjacent expression of GABA-A receptor alpha subunits

PLOS ONE

Dear Dr. Fathollahi,

Thank you for submitting your manuscript to PLOS ONE. After careful consideration, we feel that it has merit but does not fully meet PLOS ONE’s publication criteria as it currently stands. Therefore, we invite you to submit a revised version of the manuscript that addresses the points raised during the review process.

Please revise the manuscript as per comments of the reviewer.

We look forward to receiving your revised manuscript.

Kind regards,

Prasun K Datta, Ph.D

Academic Editor

PLOS ONE

Journal Requirements:

Reviewers' comments:

Reviewer's Responses to Questions

**Comments to the Author**

1. If the authors have adequately addressed your comments raised in a previous round of review and you feel that this manuscript is now acceptable for publication, you may indicate that here to bypass the “Comments to the Author” section, enter your conflict of interest statement in the “Confidential to Editor” section, and submit your "Accept" recommendation.

Reviewer #1: (No Response)

2. Is the manuscript technically sound, and do the data support the conclusions?

Reviewer #1: Yes

3. Has the statistical analysis been performed appropriately and rigorously? 

Reviewer #1: Yes

4. Have the authors made all data underlying the findings in their manuscript fully available?

Reviewer #1: Yes

5. Is the manuscript presented in an intelligible fashion and written in standard English?

Reviewer #1: Yes

6. Review Comments to the Author

Reviewer #1: Manuscript PONE-D-20-29685R1

General appointments

This is a re-review of the study of Fathollahi et al., now entitled “Unconditioned and learned morphine tolerance influence hippocampal-dependent short-term memory and the subjacent expression of GABA-A receptors alpha subunits”. Based on the previous reviewer’s criticisms the authors performed profound (and huge) changes to the article. These changes help to turn the manuscript more understandable. The study brings new and important information on the behavioral and neural events underlying morphine tolerance. However, after this second round of review, I think that the manuscript still needs to be improved to be considered for publication in PONE. In general, the manuscript lacks organization. My points to be considered by the authors are provided below:

1. The title is now precise.

2. My first criticism concerns the excessive number of abbreviations used in the manuscript. In my point of view, it is very annoying having to come back every time in the reading to remember each of the abbreviations used. This does not testify against the quality of the study but severely limited the enthusiasm of the reader. I think the authors must use only the most important acronyms, including NANT, AMT, STM, and SWM. I recommend that the authors use GABAr-�1 (or �2, and �5) instead of Gabr or Gabrs, since the word GABA itself is an acronym.

4. Another concern about the study is the absolute absence of a topic informing the total number of animals used (including losses, if any), and the number of animals by group subdivisions. In other words, how many groups were used and how many rats belonging to each group. This is important to the reader considering that any inference on the statistical analysis requires such information and this could be done through the presentation of a schematic picture of the procedures used in the study. In fact, considering the number of different dependent variables collected, this is necessary for each one of the experimental procedures used, including the STM and SWM.

5. On page 7, line 16. How long habituation to the new context lasts before saline injections?

6. Page 9, line 14. After the word “method”, insert the corresponding reference.

7. Page 18, last line, after Griesel et al., the year of publication is missing.

8. Page 19, line 23 (We observed no significant...). Looking at figure 3, It seems to me that the author’s assumption does not correlate with the information posted in the figure. If I understand the point, considering that the overall locomotion is correlated with the percentage of arm entries, both NAMT and AMT groups have reduced (and not increased) novel arm entries. Therefore, morphine decreases novel arm exploration of NAMT and AMT groups, which is reversed by valproate. Perhaps may this be due to my inability to understand the author's inference but the manuscript can largely benefit from a more organized way of discussing the data. I suggest dividing both the results and the discussion section into behavioral (morphine tolerance, short-term memory, spatial-working memory) and molecular topics.

9. Page 20, line 8 (In accordance with...). I was not able to find such data in the present report.

10. Page 21, line 14. Pervious (previous?) documents were reviewed... Reference is missing. Idem line 19 (It was noted that impairment...).

11. On page 20, line 22, the authors stated that “One of the possible mechanisms assumed to be involved in impaired memory retrieval in the Y-maze may be a manifest attenuation in synaptic performance during morphine exposure and AMT development”. Indeed, chronic exposure to morphine or heroin leads to the impairment of hippocampal LTP and induces deficits in cognitive and memory-task performance, as shown in the present study. However, the authors right after arguing that “Since morphine may abuse learning and memory circuits, a specific synaptic potentiation, which is the same as LTP, is formed in the brain”. The authors are invited to add a paragraph on this subject to discuss the learning and memory disruption in AMT rats — and the improvement following VPA, as well —, in terms of extinction impairment.

12. Graphics are now visually more understandable but progress can be made by applying a title above each figure to avoid the reader going back and forth to the manuscript. Again, I gently ask the authors to increase the size of the fonts since, in case of acceptance of the manuscript, the figures will be reduced in size. Regarding this point, In Figures 1C, 3B, and 3C, the differences between the novel and the other arm, with the start arm, could add a way to easily understand the data. It can just be done by inserting a data inset into the figures.

13. Pay attention to several typo errors and others along with the manuscript.

7. PLOS authors have the option to publish the peer review history of their article (what does this mean?). If published, this will include your full peer review and any attached files.

Reviewer #1: No

---

## [Author Response · Author response to Decision Letter 1]

7 Jun 2021

Answer to revise the PONE-D-20-29685R2

1. Please ensure that you refer to Figure 5 in your text as, if accepted, production will need this reference to link the reader to the figure.

2. 

It was edited at P17P3l 2

Answer to the reviewer comments:

General appointments

This is a re-review of the study of Fathollahi et al., now entitled “Unconditioned and learned morphine tolerance influence hippocampal-dependent short-term memory and the subjacent expression of GABA-A receptors alpha subunits”. Based on the previous reviewer’s criticisms the authors performed profound (and huge) changes to the article. These changes help to turn the manuscript more understandable. The study brings new and important information on the behavioral and neural events underlying morphine tolerance. However, after this second round of review, I think that the manuscript still needs to be improved to be considered for publication in PONE. In general, the manuscript lacks organization. My points to be considered by the authors are provided below:

Protocol DOI: http://dx.doi.org/10.17504/protocols.io.bveyn3fw

1. The title is now precise.

Thank you so much for consideration

2. My first criticism concerns the excessive number of abbreviations used in the manuscript. In my point of view, it is very annoying having to come back every time in the reading to remember each of the abbreviations used. This does not testify against the quality of the study but severely limited the enthusiasm of the reader. I think the authors must use only the most important acronyms, including NANT, AMT, STM, and SWM. I recommend that the authors use GABAr-�1 (or �2, and �5) instead of Gabr or Gabrs, since the word GABA itself is an acronym.

It was applied at the whole manuscript

4. Another concern about the study is the absolute absence of a topic informing the total number of animals used (including losses, if any), and the number of animals by group subdivisions. In other words, how many groups were used and how many rats belonging to each group. This is important to the reader considering that any inference on the statistical analysis requires such information and this could be done through the presentation of a schematic picture of the procedures used in the study. In fact, considering the number of different dependent variables collected, this is necessary for each one of the experimental procedures used, including the STM and SWM.

In some figures related to STM and SWM parameters, the number of animals inserted into the bar charts. A total number of 190 rats (including losses) were used in this study (morphine tolerance development, SWM, STM and molecular tests). It should be mentioned that different behavioral and molecular assessments were carried out on same groups of rats by the same experimenter. For evaluating the morphine tolerance development and analgesic response to VPA, 186 rats were categorized into six groups according to the following order: groups 1 and 2 (saline and VPA exposure, n = 30 for each group); groups 3 and 4 (non- associative morphine tolerance and non-associative morphine tolerance +VPA, n=36 and 30, respectively); groups 5 and 6 (associative morphine tolerance and associative morphine tolerance +VPA, n=48 and 12, respectively) (Supplementary Figures 1, 2 and Figure 1 and 2)). 60 rats were used for short term learning analyses (N= 10 rats per group) and 147 rats were used for working memory assessment (N= 30, 30, 30, 30, 12, 15 for groups 1-6, respectively (Figures 3 and 4). Finally, 36 rats were used for mRNA expression and protein level assessments (n=6 in each group). 

P6-7p4

5. On page 7, line 16. How long habituation to the new context lasts before saline injections?

It was done at P8p1l6

6. Page 9, line 14. After the word “method”, insert the corresponding reference.

It was done at the manuscript at P1p1l8

7. Page 18, last line, after Griesel et al., the year of publication is missing.

It was applied at the manuscript at P20p1l2

8. Page 19, line 23 (We observed no significant...). Looking at figure 3, It seems to me that the author’s assumption does not correlate with the information posted in the figure. If I understand the point, considering that the overall locomotion is correlated with the percentage of arm entries, both NAMT and AMT groups have reduced (and not increased) novel arm entries. Therefore, morphine decreases novel arm exploration of NAMT and AMT groups, which is reversed by valproate. Perhaps may this be due to my inability to understand the author's inference but the manuscript can largely benefit from a more organized way of discussing the data. I suggest dividing both the results and the discussion section into behavioral (morphine tolerance, short-term memory, spatial-working memory) and molecular topics.

Thank you for your consideration and accuracy. You are absolutely right and as you carefully indicated the locomotion didn’t change significantly in different groups (Fig4, and result part p17p1, line 1). We made a mistake in writing of mentioned sentences in the discussion part. In the revised version, we corrected them (Discussion part p21p2l, line 1). Also, we reviewed carefully the entire results and discussion to organize these parts better. We tried to divide the result and discussion to behavioral and molecular sub-sessions and it was applied in manuscript, since the PLOS journal does not allow to add sub-session into the discussion, we added few sentences in the first paragraph of each behavioral and molecular part of discussion.

9. Page 20, line 8 (In accordance with...). I was not able to find such data in the present report.

It was edited at P21p3l6

10. Page 21, line 14. Pervious (previous?) documents were reviewed... Reference is missing. Idem line 19 (It was noted that impairment...).

It was added at P23p3l2 and P23p4l3

11. On page 20, line 22, the authors stated that “One of the possible mechanisms assumed to be involved in impaired memory retrieval in the Y-maze may be a manifest attenuation in synaptic performance during morphine exposure and AMT development”. Indeed, chronic exposure to morphine or heroin leads to the impairment of hippocampal LTP and induces deficits in cognitive and memory-task performance, as shown in the present study. However, the authors right after arguing that “Since morphine may abuse learning and memory circuits, a specific synaptic potentiation, which is the same as LTP, is formed in the brain”. The authors are invited to add a paragraph on this subject to discuss the learning and memory disruption in AMT rats — and the improvement following VPA, as well —, in terms of extinction impairment.

Please look at p22p1-2

12. Graphics are now visually more understandable but progress can be made by applying a title above each figure to avoid the reader going back and forth to the manuscript. Again, I gently ask the authors to increase the size of the fonts since, in case of acceptance of the manuscript, the figures will be reduced in size. Regarding this point, In Figures 1C, 3B, and 3C, the differences between the novel and the other arm, with the start arm, could add a way to easily understand the data. It can just be done by inserting a data inset into the figures. 

It was applied in figures1-6 and S1-2

13. Pay attention to several typo errors and others along with the manuscript.

It was done

7. PLOS authors have the option to publish the peer review history of their article (what does this mean?). If published, this will include your full peer review and any attached files.

Do you want your identity to be public for this peer review? For information about this choice, including consent withdrawal, please see our Privacy Policy.

Reviewer #1: No

---

## [Editor Report · Decision Letter 2]

16 Jun 2021

Unconditioned and learned morphine tolerance influence hippocampal-dependent short-term memory and the subjacent expression of GABA-A receptor alpha subunits

PONE-D-20-29685R2

Dear Dr. Fathollahi,

We’re pleased to inform you that your manuscript has been judged scientifically suitable for publication and will be formally accepted for publication once it meets all outstanding technical requirements.

Kind regards,

Prasun K Datta, Ph.D

Academic Editor

PLOS ONE
---

## [Editor Report · Acceptance letter]

22 Jul 2021

PONE-D-20-29685R2 

Unconditioned and learned morphine tolerance influence hippocampal-dependent short-term memory and the subjacent expression of GABA-A receptor alpha subunits 

Dear Dr. Fathollahi:

I'm pleased to inform you that your manuscript has been deemed suitable for publication in PLOS ONE. Congratulations! Your manuscript is now with our production department. 

Kind regards, 

on behalf of

Dr. Prasun K Datta 

Academic Editor

PLOS ONE